# Autosegmentation of Prostate Zones and Cancer Regions from Biparametric Magnetic Resonance Images by Using Deep-Learning-Based Neural Networks

**DOI:** 10.3390/s21082709

**Published:** 2021-04-12

**Authors:** Chih-Ching Lai, Hsin-Kai Wang, Fu-Nien Wang, Yu-Ching Peng, Tzu-Ping Lin, Hsu-Hsia Peng, Shu-Huei Shen

**Affiliations:** 1Department of Biomedical Engineering and Environmental Sciences, National Tsing Hua University, Hsinchu 300044, Taiwan; jonas010412@gmail.com (C.-C.L.); fnwang0118@gmail.com (F.-N.W.); 2Department of Radiology, Taipei Veterans General Hospital, Taipei 112201, Taiwan; hkwang@vghtpe.gov.tw; 3School of Medicine, National Yang Ming Chiao Tung University, Taipei 112304, Taiwan; surg@vghtpe.gov.tw (Y.-C.P.); tplin@vghtpe.gov.tw (T.-P.L.); 4Department of Pathology and Laboratory Medicine, Taipei Veterans General Hospital, Taipei 112201, Taiwan; 5Department of Urology, Taipei Veterans General Hospital, Taipei 112201, Taiwan

**Keywords:** encoder–decoder architecture, DCNN, SegNet, zonal segmentation, T2W, ADC, DWI

## Abstract

The accuracy in diagnosing prostate cancer (PCa) has increased with the development of multiparametric magnetic resonance imaging (mpMRI). Biparametric magnetic resonance imaging (bpMRI) was found to have a diagnostic accuracy comparable to mpMRI in detecting PCa. However, prostate MRI assessment relies on human experts and specialized training with considerable inter-reader variability. Deep learning may be a more robust approach for prostate MRI assessment. Here we present a method for autosegmenting the prostate zone and cancer region by using SegNet, a deep convolution neural network (DCNN) model. We used PROSTATEx dataset to train the model and combined different sequences into three channels of a single image. For each subject, all slices that contained the transition zone (TZ), peripheral zone (PZ), and PCa region were selected. The datasets were produced using different combinations of images, including T2-weighted (T2W) images, diffusion-weighted images (DWI) and apparent diffusion coefficient (ADC) images. Among these groups, the T2W + DWI + ADC images exhibited the best performance with a dice similarity coefficient of 90.45% for the TZ, 70.04% for the PZ, and 52.73% for the PCa region. Image sequence analysis with a DCNN model has the potential to assist PCa diagnosis.

## 1. Introduction

Prostate disease is a major health problem in industrialized countries, especially in the Western world. One in nine men are diagnosed with prostate cancer (PCa) in their lifetime [1]. Magnetic resonance imaging (MRI) is an important diagnostic tool for PCa. The traditional diagnostic process includes measuring the level of prostate-specific antigen in the blood, followed by prostate biopsy sampling and histopathology analysis. The precision in diagnosing PCa has increased with the development of multiparametric MRI (mpMRI), which is more accurate than MRI and has become the main method for PCa diagnosis [2]. A complete mpMRI contain T2-weighted (T2W) images, diffusion-weighted images (DWIs), the corresponding apparent diffusion coefficient (ADC) map and dynamic contrast-enhanced MRI (DCE-MRI). Of them, T2W and DWI have been proved to be the most important sequences for diagnosing prostate cancer. At present, the added value DCE is not firmly established [3]. Also DCE-MRI with gadolinium chelate increases the burden on patients in terms of examination time, cost, and them potentially developing nephrogenic systemic fibrosis. A study showed that the diagnostic performance of biparametric MRI (bpMRI), which is an abbreviated prostate MRI protocol without DCE-MRI, was comparable to that of mpMRI [4]. Prostate MRI assessment relies on human experts, specialized training, and experience [5] with considerable inter-reader variability. Efforts have been made to develop a more robust approach for image assessment. This approach includes machine learning, which would help decrease the time spent on diagnosis by a considerable extent. Machine learning has thus been considered for use in the diagnosis of PCa [6].

Various segmentation methods using deep learning (DL) have been developed for automatic imaging detection and segmentation [7,8] and have been applied in oncological imaging [9,10]. Methods for medical image segmentation have gradually evolved from manual segmentation to semiautomatic segmentation and fully automatic segmentation [11]. Image recognition technology is one of the core technologies of DL. DL methods allow the simultaneous learning of adaptive image features and performance of image segmentation. In previous studies, significant achievements have been made in the use of DL for image segmentation, especially in cancer diagnosis [12,13,14].

Deep convolutional neural networks (DCNNs) are being used as a tool for image analysis. They allow for the automatic extraction of features and learning from large amounts of data for quantification. The DCNN architecture has been used for prostate segmentation or PCa detection [11,14,15,16]. In most of these studies, conventional DCNNs were used for semantic image segmentation. A common model with an encoder–decoder is the U-Net architecture, which is usually used for segmenting medical images [17]. SegNet is a modified DCNN model that uses an upsampling strategy and achieves the same accuracy as a DCNN with reduced memory and storage requirements [9]. In this paper, we propose a modified architecture based on SegNet to segment prostate zones and detect PCa through bpMRI. We compared the prediction accuracies of the autosegmentation when using different combinations of T2W images, DWIs, and ADC images of the prostate.

## 2. Materials and Methods

In this section, we discuss the method employed for training and testing convolutional neural networks for segmenting the transition zone (TZ), peripheral zone (PZ), and PCa region from T2W, DWI, and ADC MRI images. First, a simple description of the DCNN architecture is provided. Second, we describe the datasets used in this study and class labels. Third, a method is presented for dataset selection, image preprocessing, data augmentation, and training data extraction. Next, the cross-entropy loss function and training parameter used in the optimization of the network parameters are presented. We evaluated DCNN-based segmentation methods by using the following performance metrics: accuracy, dice similarity coefficient (DSC), recall, sensitivity, specificity, and the receiver operating characteristic (ROC) curve. The framework for evaluating DCNNs is shown in Figure 1.

### 2.1. Encoder–Decoder Architectures for Prostate Segmentation

A DCNN model called SegNet, which has an encoder–decoder structure, was employed in this study [18]. SegNet is a pixel-wise semantic segmentation method that was originally developed for understanding scenes and deducing the relationships among objects during autonomous driving. The architecture of encoders uses similar forward connections to that of the VGG16 architecture without fully connected layers. The encoder architecture consists of 13 convolution layers and five max-pooling indices corresponding to a set of decoders with 13 convolution layers and five upsampling layers, followed by a softmax layer for pixel-wise classification. The absence of fully connected layers is beneficial for retaining high-resolution feature maps at the deepest encoder output. Low-resolution feature maps are upsampled by SegNet by using memorized max-pooling indices. This process reduces the number of network parameters that take up memory and store boundary information in the network. A schematic of the modified SegNet architecture is displayed in Figure 2. In the conventional SegNet, a rectified linear unit (ReLU) activation function is used in the convolution block; however, we replaced ReLU with an exponential linear unit (ELU) [19]. Many experiments have been performed with different activation functions, and ELU has been found to exhibit the best performance in this architecture [20].

### 2.2. MRI Dataset

In this study, the SPIE-AAPM-NCI PROSTATEx challenge dataset was used for training, validation, and testing [21]. The PROSTATEx portal includes data from 204 patients with 330 pathologically confirmed PCa location For each patient, the coordinates of the tumor location in the MRI data were provided. The prostate MR studies were acquired on the same MRI platform. The T2W images were acquires using a turbo spin echo sequence and had a resolution of around 0.5 mm in-plane and a slice thickness of 3.6 mm. The DWI were acquired with a single-shot echo planar imaging sequence with a resolution of 2 mm in-plane and 3.6 mm slice thickness and with diffusion-encoding gradients in three directions. Three b-values were acquired (50, 400 and 800). The ADC map was calculated by the scanner software. All images were acquired without an endorectal coil. In this study, training data from 549 slices containing PCa obtained from 100 patients were selected and used in the analysis. An additional 84 slices containing PCa from 15 patients were used to test the predicted segmentation. These test data did not overlap with the training data. In order to minimize the label of imbalance, we selected the slice which contained PCa, TZ and PZ. With the tumor coordinates known, a radiologist with more than 10 years of experience labeled the PCa regions on the mpMRI images as well as the prostate anatomical zones, including the TZ and PZ. The axial T2W, DWI, and ADC images were used for comparing the accuracy of zonal segmentation with different combinations of images. The test images were selected from a dataset with known PCa locations. Each slice contained a TZ, PZ, and PCa region. A total of 80% of the images were used for training the adopted model, and 20% of the images were used for validating the model.

### 2.3. Image Processing

#### 2.3.1. Registration and Patch Extraction

To merge the DWI and ADC images with the T2W images, low-resolution DWI and ADC images were aligned and resampled by using an established registration toolbox for transformation [22]. The field of view (FOV) included the entire prostate gland, and resizing of the image of the FOV was performed by scaling the images to 256 × 256 pixels by using the nearest-neighbor interpolation method [23]. Images of different MRI sequences of the same slice location were merged. Three combinations of images were obtained from the input data: T2W images + DWIs + ADC images, T2W images + DWIs, and T2W + ADC images.

#### 2.3.2. Normalization

The image intensity is normalized to reduce the variation in the intensity distribution of images in datasets where interpatient variability exists. In this study, the normalization of images involved each pixel signal intensity subtracting the mean value of all images and dividing the obtained value by the standard deviation. Each MR sequence of a patient was normalized separately because images of different sequences are intrinsic and contain useful information for diagnosis.

#### 2.3.3. Data Augmentation

For the training sample, we augmented the data 12 times (rotation 90°, rotation −20° to 20°, horizontal flip, and vertical flip) to increase the accuracy of the adopted model.

#### 2.3.4. Class Weight Balance

In a prostate MRI scan, the anatomy of interest usually occupies a smaller region than the background, which results in an imbalance of class labels. Therefore, the background label becomes the dominant class in the learning process, which leads to imbalanced prediction. The class weighting approach was used to prevent the learning process from being trapped in local minima for improving the low dominant class [24]. More specifically, the prostate gland region was extracted from the original image and the number of pixels from the background was reduced. After the aforementioned processes, the pixel ratio was still imbalanced, with the ratio of the background:TZ:PZ:PCa zones being 58:25:14:3. To reduce the class imbalance, large weights were assigned for labels with few pixels and small weights were assigned for labels with a high number of background pixels. The pixels were classified into four classes: background, TZ, PZ, and PCa. The frequency of the classes (*F_class_*) was calculated using Equation (1). The number of pixels in a particular class is denoted as *N_class_*, and *T* represents the total number of pixels in the images. The class weight (*W_class_*) was calculated by dividing the median of *F_class_* by *F_class_*.
(1)$$F_{class}=\frac{N_{class}}{T}$$
(2)$$W_{class}=\frac{median\left(F_{class}\right)}{F_{class}}$$

### 2.4. DCNN Training

Segmentation preprocessing as well as network training, validation, and testing were performed on a single NVIDIA GeForce GTX 1070 PCIe 8 GB GPU on a Windows 10 system. The DCNN model was implemented with the Keras API (v. 2.2.4) backboned with TensorFlow (v. 1.15.0) by using Python (v. 3.6.9). We used a stochastic gradient descent optimizer to update the weights with an initial learning rate of 0.01 and a batch size of 6. Training with 50 epochs usually achieved the lowest loss and therefore was employed in our experiments. The weighted cross-entropy (WCE) loss function was used to differentiate prostate zones. This function is especially suitable when class imbalance exists. The formula for the WCE loss function is expressed as follows [25]:(3)$$\mathrm{WCE}=-\frac{1}{n}\overset{n}{\underset{i=1}{\sum }}W_{c,i}\left[T_{i}\log P_{i}+\left(1-T_{i}\right)\log\left(1-P_{i}\right)\right]$$

The summation was performed over all training images. The parameter *P_i_* is the predicted class, *T_i_* is the ground truth label, and *W_c,i_* is the class weight calculated using Equation (2) for each zone. To ensure that the trained DCNNs were stably generalized, we used five-fold cross validation. The training dataset contained 6588 slices with data augmentation. The data were split as follows: 80% for training and 20% for validation. This test process was repeated five times by using different MRI images to evaluate the full dataset. The test set consisted of 84 image slices of 15 patients. Each slice contained the TZ, PZ, and PCa regions. These slices were selected from the PROSTATEx dataset, which provided the location of PCa. The test images were grouped in a similar manner to the training data. We evaluated the performance of the zonal segmentation task of each class by using the aforementioned metrics and compared the results obtained when using T2W images + DWIs, T2W + ADC images, and T2W images + DWIs + ADC images. The DCNNs were trained and tested under three scenarios: (1) combining T2W images and DWIs into red and green channels, respectively, with the blue channel zero matrix; (2) combining the T2W and ADC images into the red and green channels, respectively, with the blue channel zero matrix; and (3) combining the T2W images, DWIs, and ADC images into the red, blue, and green channels, respectively.

### 2.5. Evaluation and Prediction Metrics

The DCNN segmentation performance was quantitatively evaluated. To compare the accuracies of true and predicted labels, we generated confusion matrices to count the number of pixels in each class. The result with a higher probability was classified as the corresponding class. The accuracy, DSC, and recall were calculated. Accuracy was defined as the ratio of the number of correctly classified pixels to the total number of pixels in the test data. The segmentation performance was also evaluated according to the intersection between label images and prediction results by using the DSC. A summary of the aforementioned three evaluation metrics is presented in Table 1. The probability map of each class was output from the softmax layer to compute the ROCs and the area under the curve (AUC) for each class. We used SciPy (version 1.1.0) for statistical computing to compare the area under the curve with the DeLong test [26] (*p* < 0.05 indicated statistical significance). The ablation study was used to show the effectiveness of loss function and activation function from our model. We selected categorical cross-entropy and WCE for loss function, and ReLu and ELU for activation function.

## 3. Results

The confusion matrices with different combinations of images are shown in Figure 3. The models trained with different combinations of images generally exhibited high performance in the background and TZ categories, followed by the PZ and PCa categories. The T2W image + DWI and T2W image + DWI + ADC image models outperformed the T2W + ADC image model in the PZ and PCa categories. This result indicates that the numbers of false negatives (FNs) in the PCa classes were higher than those in the PZ (23–39%) and TZ (15–18%) classes. Moreover, a considerable number of false positives (FPs) were found for the PZ (9–12%).

The evaluation matrices were calculated from the confusion matrices. The results obtained with each model are listed in Table 2. In terms of the DSC and recall, the segmentation of the TZ exhibited the highest performance, followed by that of the PZ and PCa region. The accuracy in segmenting the PCa region was higher than that in segmenting the TZ and PZ. However, the results for the accuracy in segmenting the PCa regions may be biased and overestimated due to the disproportionately large percentage of negative pixels among all the pixels. In this study, the scale of trues positive and true negatives was 1.78:100 for PCa. With regard to the models trained with different combinations of images, no significant difference in accuracy was observed. When segmenting the PCa regions, the DSC and recall were higher for the T2W image + DWI model and T2W image + DWI + ADC image model than for the T2W + ADC image model. The aforementioned models did not exhibit obvious differences in the DSC and recall for the TZ and PZ. The results of ablation study were shown in Table 3. The loss function and activation function for which we selected WCE and ELU respectively can obtain the optimal results of DSC for each class.

Figure 4 shows several examples of test images in the T2W image + DWI + ADC image model. The ground truth images of segmentation and the corresponding prediction results are illustrated in the aforementioned figure. The TZ in the prediction image was generally similar to that in the ground truth image. The PZ region generally was in the correct location, with considerable discordance at the boundary between true labeled image and predict image. The true PCa regions were observed in the prediction images, and FP predictions were common, especially in the PZ (9% to 12%), as indicated in the confusion matrix in Figure 3. By inspecting the original MRI images, we speculate that these FP regions were mainly formed due to the inflammation process. Moreover, in contrast to the label images, the PCa regions were usually underestimated in the prediction images, which may explain the high number of FNs of PCa in the PZ (23–39%).

The prediction results of the three adopted models, each of which was trained with a different combination of images, were compared on the basis of their discrimination ability. The ROC analysis and AUC results are shown in Figure 5. The AUCs indicate that the diagnostic performance achieved by the adopted models for the TZ was superior to that for the other zones (*p* < 0.001). The AUCs of the PCa regions in the T2W image + DWI, T2W + ADC image, and T2W image + DWI + ADC image models were 0.929 (95% CI: 0.9278, 0.9294), 0.9 (95% CI: 0.8986, 0.9006), and 0.843 (95% CI: 0.8417, 0.8441), respectively. The AUC of the T2W image + DWI model was significantly higher than that of the T2W image + DWI + ADC image model (*p* < 0.001) and T2W + ADC image model (*p* < 0.001).

## 4. Discussion

Attempts have been made to autosegment the prostate normal zonal anatomy and PCa regions by using DL methods; however, no consensus has been reached on the combination of input sequences of mpMRI images. In this study, we compared the prostate segmentation performance of models trained with different image combinations by using SegNet. We found that the three adopted models exhibited similar performance in segmenting the normal prostate zonal anatomy. The aforementioned models also exhibited similar results (DSC of 88.75–90.45%) in segmenting the TZ. These results were generally superior to those obtained when segmenting the PZ (DSC of 66.71–70.04%). The aforementioned finding has also been obtained in previous studies, which used only T2W images for training. Liu et al. [27] used the PROSTATEx dataset to perform segmentation and training by using a fully convolutional network with feature pyramid attention. They achieved a DSC of 86% in the TZ and 74% in the PZ. Aldoj et al. [28] also used T2W images of the PROSTATEx dataset. They used a DenseNet-like U-Net for training and achieved a DSC of 89.5% ± 2% in the central gland region and 78.1% ± 2.5% in the PZ. Khan et al. [16] used a model resembling the encoder–decoder architecture and only T2W images of two prostate MRI datasets for segmentation and training. They achieved a DSC of 90.8 ± 1.2% in the central gland region and 76.0 ± 3.9% in the PZ by using SegNet [16]. The results of the present study indicate that SegNet achieved a comparable performance in the segmentation of the normal prostate zonal anatomy irrespective of the training datasets and number of training samples. SegNet uses a small amount of memory and stores boundary information in the network. As demonstrated in previous studies, DL methods cannot suitably discern the boundaries of the PZ. Models combining different sequences do not exhibit superior results to models that employ T2W images alone. When segmenting the PZ, the performance of the T2W image + DWI model was similar to that of the T2W image + DWI + ADC image model and marginally higher than that of the T2W + ADC image model.

Previous studies have attempted the autosegmentation of PCa zones by using only T2W images [15] or DWIs [29]. Although these studies have exhibited suitable performance for PCa segmentation, a mono model with a single-sequence input provides limited information about the prostate and may provide different results when applied to different datasets. Some studies have demonstrated the advantages of using functional sequences for PCa autosegmentation. Liu et al. made selections from a set of 87 lesion features to determine the model ensemble for XmasNet [30]. They found that frequently appearing features are usually related to functional information, including correlation, variance, the minimum intensity of the lesion region on the Ktrans image, and the minimum intensity of the lesion region on an ADC map. Song et al. found that for combinations of different imaging sequences, the models trained with DWIs exhibited higher specificity than the other models did. The features of PCa regions with a high signal intensity in DWIs are learned more easily by a DCNN than by models that include only T2W + ADC images [16]. Our study used different combinations of images and found that models containing DWIs exhibited the best performance. The performance of the T2W image + DWI + ADC image model (DSC of 52.73%) and T2W image + DWI model (DSC of 51.92%) was significantly higher than that of the T2W + ADC image model (DSC of 36.62%). As indicated by the confusion matrices, the T2W + ADC image model had a higher number of FPs and FNs than did the models with DWIs. DWIs have been used in prostate MRI and are key in the prostate mpMRI exams [31,32,33]. DWIs should include high-b-value images and the corresponding ADC map. A high-b-value DWI indicates the preservation of the signal in areas of restricted or impeded diffusion and thus highlights tumors. A diminished signal indicates normal tissue. On an ADC map, a low ADC value indicates that diffusion is restricted and anatomical information is preserved. Compared with cases in which ADC maps are used alone, the discernability of clinically significant cancers is sometimes improved in high-b-value images, especially those captured adjacent to the anterior fibromuscular stroma, in a subcapsular location, and at the apex and base of the gland [3]. The results of this study indicate that although ADC maps can provide PCa diagnostic information, their performance in PCa detection is lower than that when using DWIs. 

In this study, we used SegNet and achieved comparable or superior results for PCa autosegmentation (DSC of 0.52) to those achieved in other studies using bpMRI or mpMRI (DSCs of 0.37–0.46) [34,35,36]. Our study achieved a result (AUC of 0.9) comparable to that achieved by other studies (AUCs of 0.84 [30] and 0.94 [14]) that used the AUC to determine the performance of PCa detection for the dataset and combination of images adopted in the present study. The comparison of the network with state-of-the-art methods was show in Table 4. SegNet occupies a smaller computing space and is faster than other architectures. Moreover, it does not require multistage training. To maintain the same output image resolution the same as the input image resolution, SegNet uses pooling indices to upsample low-resolution feature maps. This process allows the network to store boundary information and reduce the number of trainable parameters. We also employed image augmentation and patch extraction to reduce the amount of training data while achieving similar or superior results to those obtained in previous studies.

Several challenges exist in the DL process for PCa autosegmentation. The first challenge is the imbalanced class labels. The PCa regions are usually smaller than those of background normal tissue, which causes the background label to become the dominant class during the learning process and thus unbalance prediction. To avoid the learning process from being trapped in a local minim, we used the class weight balance method to increase the weighting of the small class and increase its prediction accuracy. The accuracy can be further increased by increasing the number of datasets and using class-distribution-aware training techniques [18]. Another challenge is the high FP rate of PCa in the PZ region. By inspecting the source images, we found that this phenomenon may be due to the inflammation process, frequently encountered motion artifacts, or rectal gas causing susceptibility artifacts in DWIs [37]. With nearly 75% of malignant lesions emerging from the PZ [38], improving the imaging quality and decreasing artifacts are essential for improving the DL performance.

Our study has some limitations. First, the size of our training data might be suboptimal for DL. Second, the training parameters in our model are yet to be improved, especially for the autosegmentation of the normal PZ and PCa region. A fine-tuning strategy may help train the model and improve the segmentation performance. Third, the training was performed using a dataset with homogeneous imaging quality by the same vendor. We have yet to validate whether the model is sufficiently robust or applicable to different datasets.

## 5. Conclusions

We developed a strategy for autosegmenting prostate MRI images. The developed strategy involves using SegNet and conducting image preprocessing (by combining different image sequences obtained through bpMRI), prostate extraction, training data augmentation and class weight balancing. The developed strategy can be an efficient method for identifying the TZ, PZ, and PCa regions with satisfactory performance. DWIs are an essential component that should be included in the training model for detecting the PCa region.

## Figures and Tables

**Figure 1 sensors-21-02709-f001:**
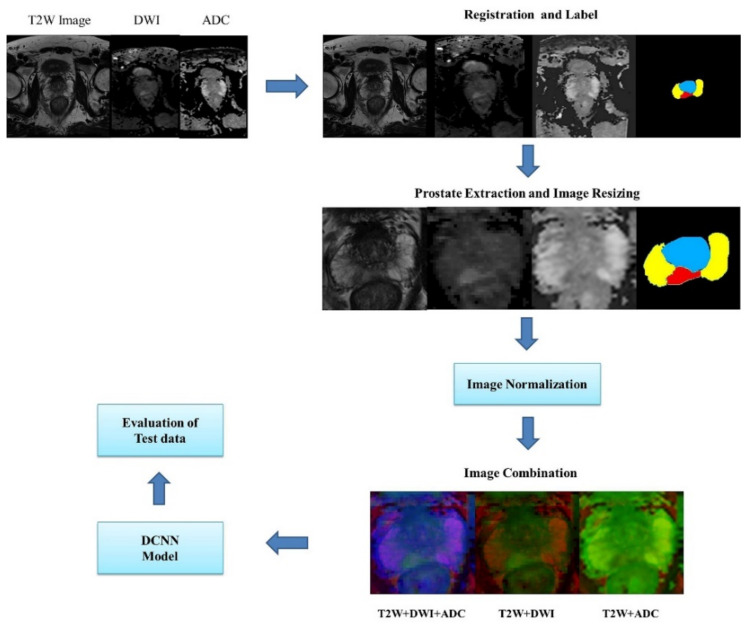
Method for evaluating encoder–decoder DCNN architectures in the semantic segmentation of prostate bpMRI images.

**Figure 2 sensors-21-02709-f002:**
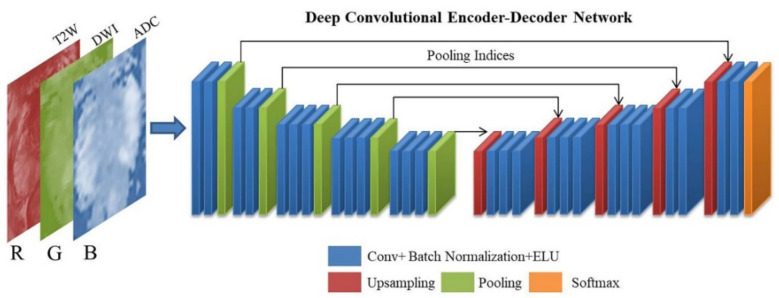
Schematic of the modified SegNet architecture.

**Figure 3 sensors-21-02709-f003:**
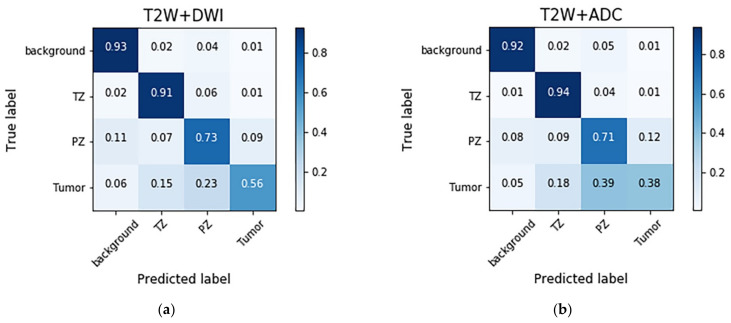
Normalized confusion matrices for the prediction results obtained with different image combinations. The results are presented in percentages. The total number of pixels was 5,570,560 (256 × 256 × 85). (**a**) T2W image + DWI model, (**b**) T2W +ADC image model, and (**c**) T2W image + DWI + ADC image model.

**Figure 4 sensors-21-02709-f004:**
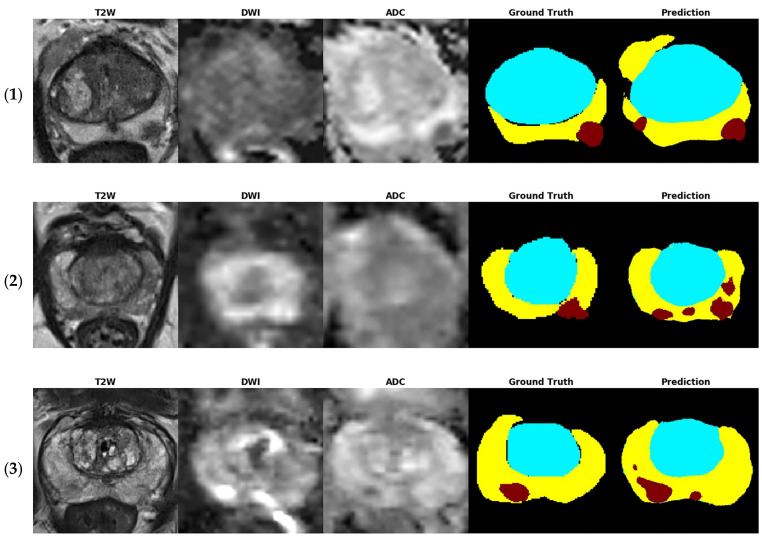
Test images of five patients (**1**–**5**) in the T2W image + DWI + ADC image model. The tumor location was provided from dataset. The ground truth images of segmentation and the corresponding prediction results are illustrated (blue region: TZ, yellow region: PZ, and red region: PCa).

**Figure 5 sensors-21-02709-f005:**
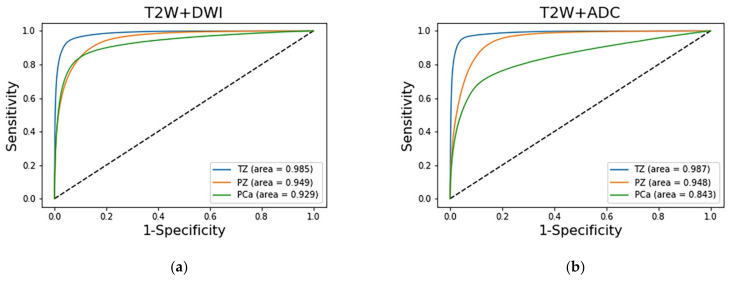
Comparison of the ROC curves of the three prediction models for discriminating among the TZ, PZ, and PCa region. The blue, orange, and green lines represent the predictions for the TZ, PZ, and PCa region, respectively. Different combination of the (**a**) T2W + DWI. (**b**) T2W + ADC. (**c**) T2W + DWI + ADC.

**Table 1 sensors-21-02709-t001:** Definitions of the evaluation metrics.

Metric	Formula
Accuracy	$\frac{N_{TP}+N_{TN}}{N_{TP}+N_{TN}+N_{FP}+N_{FN}}$
DSC	$\frac{2N_{TP}}{2N_{TP}+N_{FP}+N_{FN}}$
Recall	$\frac{N_{TP}}{N_{TP}+N_{FN}}$

Note: *N_TP_* = number of true positives, *N_TN_* = number of true negatives, *N_FP_* = number of false positives, *N_FN_* = number of false negatives.

**Table 2 sensors-21-02709-t002:** Results of multiclass segmentation on different combination of SegNet-like models.

	Accuracy			DSC			Recall		
	TZ	PZ	PCa	TZ	PZ	PCa	TZ	PZ	PCa
T2W+DWI	95.2	92.1	96.93	88.75	68.93	51.92	0.91	0.73	0.56
T2W+ADC	95.73	91.48	96.07	90.22	66.71	36.62	0.94	0.71	0.38
T2W+DWI+ADC	95.87	92.38	96.97	90.45	70.04	52.73	0.93	0.74	0.57

The accuracy and DSC are presented in percentages.

**Table 3 sensors-21-02709-t003:** The results of ablation study in Terms of the Loss functions and activation function.

Parameter	Accuracy			DSC			Recall		
	TZ	PZ	PCa	TZ	PZ	PCa	TZ	PZ	PCa
ReLu+CC	94.57	90.37	97.3	87.2	63.72	36.49	0.88	0.7	0.26
ELU+CC	94.52	91.1	0	87.6	66.49	0	0.92	0.73	0
ReLu+WCE	94.16	88.59	96.18	85.15	61.78	46.63	0.8	**0.77**	0.56
ELU+WCE	**95.87**	**92.38**	96.97	**90.45**	**70.04**	**52.73**	**0.93**	0.74	**0.57**

CC = loss function of categorical cross-entropy, WCE = loss function of weighted cross-entropy, Bold values indicate best parameters.

**Table 4 sensors-21-02709-t004:** Performance comparison of the network with state-of-the-art methods.

Reference	Methods	DSC			AUC
		TZ	PZ	PCa	PCa
Khan et al. [16]	SegNet	90.8 ± 1.2	76.0 ± 3.9	-	-
Liu et al. [27]	Fully convolutional network	86	74	-	-
Aldoj et al. [28]	DenseNet-like U-Net	89.5 ± 2	78.1 ± 2.5	-	-
Simon et al. [34]	Fully convolutional network	-	-	41	-
Coen de Vente [35]	U-Net	-	-	37.46	-
Yusuf [36]	cost-sensitive support vector machines (SVMs)	-	-	46	-
Song, Y. [14]	VGG-Net	-	-	-	0.94
Liu S. et al. [30]	XmasNet	-	-	-	0.84
T2W + DWI + ADC (Our result)	Modified-SegNet	90.45	70.04	52.73	0.9

The DSC is presented in percentages.

## Data Availability

Data is contained within the article. The data presented in this study are available in article.

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
