# Peer review of "Autosegmentation of Prostate Zones and Cancer Regions from Biparametric Magnetic Resonance Images by Using Deep-Learning-Based Neural Networks"

_sensors, 2021, doi:10.3390/s21082709_

Round 1

Reviewer 1 Report

This article describes a method for autosegmenting the prostate zone and cancer region by using SegNet, a deep convolution neural network (DCNN) model.  The AU’s approach utilizes T2-weighted images, diffusion-weighted images, and apparent diffusion coefficient images.  These images exhibited the best performance and the model has the potential to assist prostate cancer diagnosis.

The study is appropriate for the journal, but a few clarifications are necessary. 

The study uses only images with prostate cancer.  Shouldn’t images of normals have been included?  The AU’s state this in the last paragraph of the discussion, but perhaps this should be explained earlier also.

Table 2 needs a title.  (line 245)

Weighted magnetic resonance images is a general term.  Just because an image has that title does not imply that the acquisition parameters are the same, even if they were acquired on the same MRI platform.  What were the acquisition parameters for the various weighted images?

It seems as though diffusion-weighted images and apparent diffusion coefficient images would provide redundant information.  What are the other dependencies in the weighted images (T1? spin density? etc). Were other images tried and if so which ones?

In 1993 Fletcher (Magn Reson Med 29:623-630) demonstrated utility of using pure T1, T2, and spin density magnetic resonance images for unsupervised segmentation of brain tissues.  This approach was superior to utilizing weighted images.  Why were weighted images utilized in this study?  Would the processing have been less and the segmentation better if pure T2 and diffusion coefficient images were used? 

Reviewer 2 Report

Authors propose modified SegNet as segmentation network to segment prostate zones and detect PCa through bpMRI. The improvement of modified SegNet is replacing the ReLU with ELU activation function. The method is evaluated on PROSTATEx.

There are the major problems as follow.

  1. There are a lot of segmentation methods but authors mention only few. Example of segmentation methods, 10.1109/TPAMI.2018.2844175, 10.1109/TPAMI.2016.2572683 2. SegNet is old semantic segmentation for scene understanding, while U-Net is proposed for Medical image segmentation. 
  2. Also authors mention in page 2 line 64 that "A common model with an encoder–decoder is the U-Net architecture, which is usually used for segmenting medical images" So, why choose SegNet as segmentation network instead of U-Net. It is better to show the comparison between U-Net and SegNet.
  3. The evaluation is slice level, volume level, or patient level?
  4. Authors not compare with existing works. Authors should provide state-of-the-art section to compare the performance of proposed network.
  5. Authors may consider to add an ablation studies to show the effectiveness of ELU, and loss function.

Round 2

Reviewer 2 Report

Overall, I am quite satisfied with the response given by the authors, and glad to see that the quality of the paper has been improved substantially. The experiment in this revised version is more extensive and constructive than the last one. I think the paper now is more likely to be published.